# Meaning-Making among Drug Addicts during Drug Addiction Recovery from the Perspective of the Meaning-Making Model

**Tri Iswardani [1,\*], Zahrasari Lukita Dewi [2], Winarini Wilman Mansoer [1] and Irwanto Irwanto [2]**

[1]  Faculty of Psychology, University of Indonesia, Jl. Lkr Kampus Raya—Prof. Dr. R. Slamet Iman Santoso, Pondok Cina, Kecamatan Beji, Kota Depok 16242, West Java, Indonesia

[2]  Faculty of Psychology, Atma Jaya Catholic University of Indonesia, Jl. Jend. Sudirman No 51, Karet Semanggi, Setiabudi, South Jakarta City 12930, Jakarta, Indonesia

\*  Correspondence: dani-tri@ui.ac.id or danisadatun@gmail.com; Tel.: +62-217270004

**Abstract:** (1) Background: This study aimed to explore the dynamics of meaning-making among drug addicts during drug addiction recovery by using a case study approach. The participants were five male recovering addicts, aged 26 to 49 years, who had been abstinent for 4 to 17 years; (2) Methods: Data collection consisted of in-depth interviews. Data were analyzed using the Adverse Childhood Experience (ACE) Questionnaire, Meaning in Life Questionnaire (MLQ), Beck's Depression Inventory-II (BDI-II), and recovery stage criteria based on the Developmental Model of Recovery (DMR). (3) Results: The results showed that meaning making was an ongoing process before and during the use of drugs and recovery. Drug use was a coping strategy to overcome distress caused by ACE, which was perceived as a stressful event and brought up an implicit meaning of ACE. This initial meaning played an essential role in initiating the dynamics of the subsequent meaning-making process. A similar mechanism using non-constructive adaptation processes other negative impacts of drug use. They created more distress and developed false-positive beliefs, which resulted in continued drug use. Symptoms of depression occur during drug use, which drags the addict to the lowest point in life (hitting rock bottom), and addicts perceive it as a turning point for seeking treatment and attaining recovery. In complete recovery, reappraising the meaning of the stressors experienced throughout life makes new constructive meaning. Creating a constructive meaning of earliest traumatic experiences played an important role in preventing relapse and ensuring the success of recovery from drug addiction.

**Keywords:** adverse childhood experience; depression; drug addiction; meaning in life; recovery

## 1. Introduction

Drug addiction affects all aspects of the addict's life. A whole person's recovery from drug addiction should address all aspects of the addict's life. Gorski [1] proposed the concept of the Developmental Model of Recovery (DMR), which is a model that explains that recovery from addiction follows a pattern of certain stages with a specific task to complete. The bio-psycho-social-spiritual model (BPSS) is an approach that covers all aspects of addicts' life. Including "spirituality" in this BPSS model has opened new horizons for understanding drug addiction recovery. It emphasizes the importance of spiritual aspects to complement recovery in biological, psychological, and social factors. In particular, the spiritual element explains more about the reasons for the life of addicts and how they redefine their meaning in life. In line with the BPSS model, the humanistic-existential perspective explains that drug addiction is an existential dilemma [2].

Moreover, drug addiction is also a response to a life that lacks personal meaning caused by the addict's hampered ability to choose an authentic and meaningful life [3,4]. The addict's negative perception of life may cause this condition [5]. The recovery process from the spiritual perspective includes finding one's identity, understanding oneself, seeking a

relationship with God, gaining a sense of meaning, and searching for meaning [6]. Kemp [7] asserted that addiction to anything inflicts considerable damage to a person's sense of self, which needs to be reconstructed during recovery until the addict retrieves a sense of meaning in life.

In other words, to overcome drug addiction, some key questions need to be addressed, including "Why did I become a drug addict?", "What was I looking for by taking drugs?", "What did I get by using drugs?", "What meaning of life have I found from using drugs?". These questions reflect that addicts experience an existential problem.

Given that, to live a clean and sober life, addicts need to find reasons to live a more purposeful life. Meanwhile, many spiritual-based rehabilitation programs in Indonesia focus more on religion as a vertical dimension of spirituality. According to Koenig and Puchalski et al. [8,9], apart from being seen as a sacred or transcendent relationship with God (vertical spiritual dimension), spirituality also implies a personal search to understand answers to questions about the meaning of life (horizontal spiritual dimension). The spiritual dimension may (or may not) lead to religiosity.

The above discussion shows that drug addiction recovery can be seen as a way to overcome a loss of self-identity and reconcile one's authentic spiritual self with their biological, psychological, and social self. To support the implementation of the BPSS approach in drug addiction recovery programs in Indonesia, we contend that the programs can complement a spiritual-based framework that emphasizes the importance of having meaning in life.

Meaning in life relates to things considered essential and valuable, giving specific values to someone and ultimately giving a person's life purpose. In other words, meaning and happiness in life occur when individuals have a sense of purpose and succeed in fulfilling those life goals [3]. Meaning in life is also often associated with an understanding that the world and that life is something that has coherence [10].

In the context of drug addiction recovery, Puentes [11] conducted a mixed-method study on the search for meaning, presence of meaning, meaning-making framework, and fulfillment of meaning in life among addicts in rehabilitation. The result found that meaning in life correlates with psychopathology, psychological well-being, coping, and motivation to use drugs. More specifically, fulfillment of meaning had the most substantial impact on psychological well-being, whereas searching for meaning correlates with psychological distress. Meanwhile, by implementing the Meaning in Life Questionnaire (MLQ), Posson [12] found two significant differences between searching for meaning and the presence of meaning in different recovery stages among drug addicts. Furthermore, Chen [13] explained that self-compassion—a strategy to manage emotions in which negative experiences are seen or accepted with consciousness, goodwill, understanding, and a sense of humanity—plays an essential role in drug addiction recovery.

George and Park [14] defined meaning-making as a process in which someone sees their life as something that has coherence, is directed, motivated by valuable purposes, and is essential in the world. Meaning in life links to lower distress and impairment, greater well-being and achievement, and these findings also extend to physical health [15,16]. Steger [16] suggested the existence of the Presence of Meaning and the Searching of Meaning. Meanwhile, Park [17] stated that there is a Meaning-Making process and the Meaning-Made. These two theories have similarities in understanding the process of finding the meaning of life. The Meaning-Making process is similar to the Searching for Meaning, whereas the meaning Made is similar to the Presence of Meaning. Park [17] in the theoretical framework called the Meaning Making Model (Figure 1) developed the dynamics of meaning-making in a structured manner. In the context of understanding meaning in life, the model covers the following principles: (1) an individual has a set of orienting systems called global meaning (GM)—the cognitive frameworks to help interpret life experiences and provide motivation; (2) when being faced with a situation that has the potential to cause stress, an individual will appraise the situation and make meaning of it, with the resultant meaning called "appraised situational meaning" (ASM); (3) discrepancies between ASM

and GM will determine the extent of the distress; (4) the experienced distress will initiate the process of meaning-making; (5) through the process of meaning-making, an individual reduces discrepancies that have caused distress and restores their negative perception to enable them to see that the world and their existence are worthwhile; (6) if the process of meaning-making is successful, an individual may adapt themselves to the stressful event (they have produced meaning made).

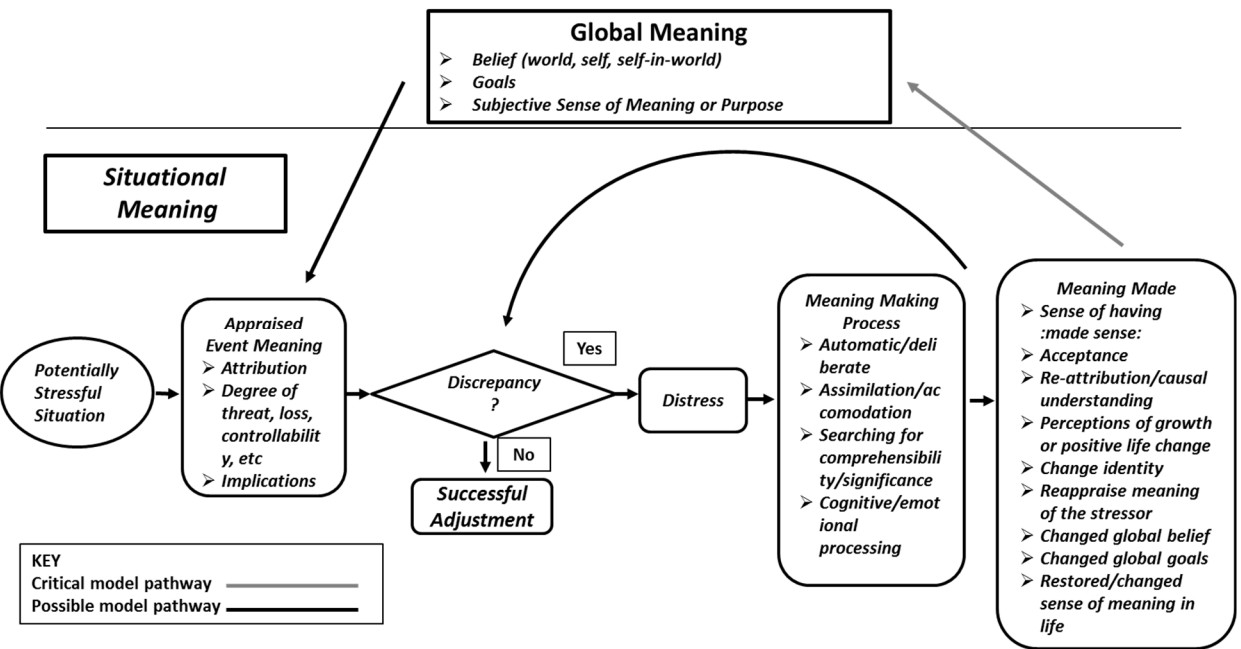

**Figure 1.** The meaning making model ([17], p. 258).

Meaning made is always positive in terms of its role in helping individuals to enhance well-being. According to Park [17], until meaning-making attempts result in some change that reduces the discrepancy between appraised and global meaning, they may be positively related to distress; over time, meanings made (and concomitant decreases in discrepancies) should be related to better adjustment. Thus, attempting to make meaning is not necessarily linked with adjustment but may merely signal an ongoing discrepancy between an individual's global meaning and an event's appraised meaning. In addition to resolving discrepancies by changing the appraised meaning of stressors, individuals can make changes in their global meaning [17]. For example, global belief changes may involve coming to see the illusion of safety in life, after experiencing a natural disaster. There are many instances of people experiencing losses. For example, a mother who lost her beloved child changed her global beliefs from seeing God as being benevolent to God as being unfair, and after going through deep sadness and grief, she changed back to her belief that God is merciful and loves her child more than anyone else, and finally believes that her child is in better hands. Meaning making can also result in identifying goals that are not attainable and abandoning them or substituting alternative goals. A young man who was eager to continue his education to achieve university from his belief that higher education was the only way to gain respect from people decided to find a job because he could not afford to pay for college and changed his view that success in life is not solely by taking higher education, but economic independence.

As a process that occurs throughout life, meaning making is a continuous effort in the form of deep contemplation of disturbing thoughts and long-term distress. Through the process of meaning-making, an individual will produce meaning made. The specific stages are as follows: (1) feeling that something makes sense; (2) there is acceptance; (3) doing re-attribution/having a causal understanding; (4) there is a perception of growth or a positive change in life; (5) changing an identity; (6) reappraising meaning of the

stressors; (7) changing global beliefs; (8) changing global goals; and (9) restoring/changing a sense of meaning in life [17].

The Meaning Making Model sees recovery from a stressful event as an automatic process and conscious coping activity, similar to recovery from trauma [17–19]. Park [20] found that individuals who have experienced trauma tend to change their global goals by devoting their lives to the fields related to the trauma they have experienced. Joseph and Linley [21] found that individuals who had experienced trauma and then post-traumatic growth (PTG) made accommodations through meaning-making that changed their global beliefs or goals. For example, after attaining PTG, a rape victim went through the process of meaning-making using accommodation by devoting her life as a volunteer in an NGO specializing in recovering other rape victims. Beliefs she was a weak victim and transformed into beliefs as a strong and capable person in helping others. In the context of drug addiction, during fieldwork, we found that many former drug addicts devoted their lives to being addiction counselors to help individuals recover from drug addiction. Meanwhile, Chen [22] found that drug addicts might experience distress several times which caused suffering, but the suffering might become a strong motivation for recovery. Hicks and Routledge [6] explained that distress might indirectly open up the opportunity for an individual to find new sources of meaning to replace their lost meaning in life by searching for a new meaning.

Potential stressful situations causing distress that initiate the meaning-making process can come in various forms, including Adverse Childhood Experiences (ACE). There are ten types of ACE: emotional abuse, physical abuse, sexual abuse, emotional neglect, physical neglect, parental divorce, witnessing violence in the family, substance abuse in the family, mental illness in the family, and imprisoned family members [23]. Research has found that children exposed to four or more types of ACE during the first 18 years of life are 11 times more likely to become drug users; those with six or more are 46 times more likely to be drug users with a needle [24]. Regarding drug addiction recovery, almost all addicts experience a traumatic event [25]. Concerning addiction recovery, addicts who experienced trauma would have more incredible difficulty in maintaining recovery, and conversely, they would also have more difficulty recovering from trauma [26]. The facts above indicate that the Meaning Making Model is a framework that is suitable for explaining the dynamics of acquiring meaning in life. This qualitative case study aimed to investigate the process of meaning-making before and during drug use and recovery among former drug addicts. The question of the study is as follows: How was the process of meaning-making among drug addicts before they used drugs, while they were using drugs, and while undergoing recovery?

## 2. Methods

### 2.1. Participants' Criteria, Selection, and Data Collection Method

This study was conducted in 2020 in Jakarta. The criteria used in the selection of the participant include (a) the individuals have stopped using drugs (recovering addict) for at least three years; (b) the individuals are willing to undergo an in-depth interview regarding addiction experiences; and (c) the individuals stated that they are able and willing to participate in this research.

Participants in this study had participated in the previous study on the relationship between adverse childhood experiences (ACE), depression, meaning in life, and drug addiction, which involved 251 lifetime drug users. We selected nine recovering addict participants based on the defined criteria. The first initial interview, applied as a screening stage, was conducted in person and lasted 40–60 min, followed by further in-depth interviews using a video conferencing platform for 60–90 min. After completing the transcription, the researcher carried out a follow-up session using telephone calls for 10–20 min to reconfirm that the data provided by the participants was under the researcher's understanding. From this stage, we only found 5 participants committed to continuing and participating in this study by signing written informed consent.

Considering the theme of the case study is personal and sensitive, it is essential to build a good relationship with the participants. Therefore, the researcher conducted face-to-face screening (initial) interviews to carefully identify potential participants who met the predetermined selection criteria and candidates who could best articulate their experiences and re-examine participants' responses to ACE. Due to the COVID-19 pandemic, the online interview was the best-fitted, safest, and most possible technique to continue the qualitative data collection process.

This study used descriptive data on the meaning of life and depressive symptoms from the previous study to ensure the validity of participants' responses.

Participants chose pseudonym initials to protect their identity. The researcher recorded the entire interview process in audiotaped and video form for further transcription. The interviewer has a Master's degree in mental health, was familiar with working with addiction clients, and could approach participants with natural curiosity and respect to ensure honest answers from participants. An independent professional verbatim transcriber with a Bachelor's degree in psychology participated in this study.

### 2.2. Design

This study used a qualitative research design to explore recovering addicts' life experiences. This research design is applied to seek empirical knowledge through in-depth investigation and analysis of a phenomenon in real life using certain theoretical concepts [27]. This current study is also a theory-building case study research to enrich the Meaning Making Model invented by Park [18]. Theory-building case studies refine, enrich, or modify existing theories by comparing theory with the observations in a particular case [28]. Specifically, this study focuses on the meaning-making process in recovering addicts' life experiences, starting from childhood until the occurrence of drug addiction, its discontinuance, and eventually reaching recovery.

### 2.3. Procedure

The interviewer captured the meaning-making process, starting from the participants' experiences in childhood, during the occurrence of drug addiction in the participants, and until the participants reached recovery, using a semi-structured interview guide (Table 1).

**Table 1.** Semi-structure interview guide.

| | |
|---|---|
| 1 | Describe life experience as a person who experienced drug addiction from childhood to the present. |
| 2 | What is the meaning of drugs and drug use throughout life experience as a recovering addict? |
| 3 | Possible adverse experiences in the first 18 years and other life crises throughout life? |
| 4 | Describe the impact of drug use (belief about self, emotions) |
| 5 | Describe the life meaning before using drugs, while using, and after reaching recovery |
| 6 | Define the meaning in life as recovering addict. |

Based on the participants' answers to the questions using the interview guidelines above, the next step is to carry out verbatim transcription involving three data analysts, then mark the essential phrases related to the meaning-making process. Data analysts then identified and reviewed the emerging themes and general patterns, then put them in the form of a report. Three analysts carried out inter-rater reliability.

At the end of the interview, participants were debriefed and given information about the follow-up treatments option should the questions raise disturbances, knowing that the subject of the interview was sensitive to re-experiencing traumas. No one reported disturbances after completing the study. Most participants appreciated that answering questions raised in the interview gained insight into their life history. Participants received an internet quota voucher worth USD 20 in exchange for transportation and the internet quota used for online communication.

*2.4. Data Analysis*

Deductive thematic analysis was applied to obtain information on the participants' experiences before and during the drug use period, after stopping drug use, and during recovery. A deductive approach involves coming to the data with some preconceived themes expected to find reflected there, based on theory or existing knowledge [29]. The Meaning Making Model [17] was used as the main framework to analyze data. Data were interpreted using content analysis to identify the patterns and themes in the investigated cases [28]. Additional analysis using the Developmental Model of Recovery (DMR) [1] was taken to identify participants' stages of recovery at the time of this study, which consists of six stages with their respective characteristics, namely: (1) Transition; (2) Stabilization; (3) Early Recovery; (4) Middle Recovery; (5) Late Recovery; and (6) Maintenance.

Based on the descriptions provided by the participants, the researcher identified several constructs in the Meaning Making Model, including potentially stressful situations, the appraised meaning of events, alignment of discrepancies between global meaning and situational meaning, distress, types of meaning-making processes, and meaning made throughout the experience of addiction and recovery.

## 3. Results

*3.1. Demographic Characteristics*

Table 2 below shows participant data by age, gender, initial use, length of drug use, and recovery period. Five male participants (AG, BB, MA, PR, SG) were aged 24–49. Participants had experienced drug use for 10–17 years and had stopped using drugs or abstained entirely for 4–19 years. The five participants had undergone a drug rehabilitation program. The educational background of the participants ranged from high school graduates to undergraduates. One participant was diagnosed with HIV, and the other had a nephrectomy. When the study was conducted, all participants reported being in good health.

**Table 2.** Demographic characteristics and drug use.

| Participant | Age (Year) | Gender | Initial Use (Age) | Length of Drug Use (Year) | Recovery Period (Year) |
|---|---|---|---|---|---|
| 1. AG | 37 | male | 10 | 10 | 17 |
| 2. BB | 26 | male | 12 | 10 | 4 |
| 3. MA | 41 | male | 10 | 15 | 16 |
| 4. PR | 49 | male | 13 | 17 | 19 |
| 5. SG | 27 | male | 12 | 11 | 4 |

*3.2. Descriptive Results of ACE, Meaning in Life, and Depression Symptoms*

In this study, we used several measurements as data triangulation, consisting of the Adverse Childhood Experience Questionnaire (ACE) [30], Meaning in Life Questionnaire (MLQ) [16], and Beck's Depression Symptoms (BDI)-II [31].

The results of the descriptive analysis in Table 3 below show that participants experienced adverse childhood experiences ranging from 1 to 8. Within this range of ACE scores, emotional neglect was the type of ACE experienced by all participants. Other types of ACE occur in variation. One participant went through almost all types of ACE. (see also Table 4). All possible variations are found based on two dimensions of meaning in life, namely the presence of meaning and searching for meaning, categorized into high and low.

Three participants experienced symptoms of depression categorized as normal, but one showed mild mood disorders, and the other showed borderline clinical depression.

**Table 3.** Descriptive results of ACE, meaning in life, and depression symptoms.

| Participant | ACE Score | Meaning in Life (Presence/Search) | Depression Symptoms |
|---|---|---|---|
| 1. AG | 4 | 19/25 (Low/High) | 5, Normal |
| 2. BB | 3 | 14/22 (Low/Low) | 15, Mild mood disturbance |
| 3. MA | 3 | 34/17 (High/Low) | 6, Normal |
| 4. PR | 1 | 29/32 (High/High) | 4, Normal |
| 5. SG | 8 | 30/20 (High/Low) | 20, Borderline clinical |

**Table 4.** ACEs and other life crises before drug use.

| Identities of the Participants | | | | |
|---|---|---|---|---|
| AG | BB | MA | PR | SG |
| 37 years; 17-year recovery period | 27 years; 4-year recovery period | 41 years; 16-year recovery period | 49 years; 19-year recovery period | 24 years; 4-year recovery period |
| **ACEs, drug use, and other life crises** | | | | |
| ACE = 4<br>1. Emotional and social neglect<br>2. Verbal and physical abuses<br>3. Loss of a parent<br>4. Living together with a drug addict at home | ACE = 3<br>1. Verbal abuse<br>2. Sexual abuse<br>3. Emotional neglect | ACE = 3<br>1. Emotional neglect<br>2. Witnessing violence committed by parents<br>3. Parent's separation | ACE = 1<br>1. Emotional neglect | ACE = 8<br>1. Physical abuse<br>2. Verbal abuse<br>3. Loss of a parent<br>4. Father being a drug addict<br>5. Father being incarcerated<br>6. Physical neglect<br>7. Emotional neglect<br>8. Witnessing violence at home |
| **Drugs abuse consequences and other life crises** | | | | |
| -Bullying at school<br>-Motor accident<br>-Expelled from school several times | -HIV<br>-Arrested by the police | -Divorce<br>-Embezzlement<br>-Nephrectomy | -Divorce<br>-Mother was seriously ill and then died<br>-Arrested by the police | -Being adopted by an uncle<br>-Being separated from brothers and sisters |

### 3.3. Family Background and Drug Addiction History

Participants' backgrounds include parenting patterns, birth order, parental attachment relationships, and social interaction. The history of drug addiction includes initial use, lengths of use, and recovery period achieved.

Biological parents did not raise two participants. AG is an only child and was raised by his grandparents. He never knew his father, who left his mother before he was born. He was raised in an authoritarian manner and with strict discipline. BB is the eldest of four children. His relationship with his temperamental father, who had passed away two years ago, was never close. MA is the eldest boy of two siblings. He was raised by multiple caretakers, including grandparents, uncles, aunts, and babysitters. However, practically, MA did not have enough supervision from the caretakers. His mother died when he was 35 years old. PR is the eldest of two siblings. He was raised by hard-working parents who spoil and pamper their children with excellent facilities and luxuries. His mother died when he was 33 years old. SG is the third of four children from his biological family and the only child in an adopted family. His grandparents raised him from the age of 3–6 years. His biological mother left her children when SG was six years old. SG has not seen his father since he was five years old. His biological father was a drug addict and drug dealer who had been incarcerated for drug trafficking. SG's father also frequently abused physically and verbally towards SG's mother and siblings. His authoritarian uncle adopted SG at the age of 7 years.

The history of drug use of participants of this study started in grade 4 elementary school to grade 2 junior high school. Their initial motives are the desire to try drugs and to be accepted to associate with affiliate groups. This motive arises with the reason to fulfill the need to feel pleasure for a moment. However, in the end, they continued to use drugs more frequently (intensively), marked by a change in the type of drug used, from mild drugs to hard drugs with more potentially addictive effects. Peer acceptance was the primary motive for all participants, and this caused the recreational use to increase to the intensive use stage. Drug use is a means to increase self-confidence and provide a calmer and less impulsive state. The satisfaction obtained when using drugs is not only from the effects of drugs but also the pleasure of gathering to use drugs together.

Teachers and school officials recognized their recreational and intensive drug use in all cases. As a consequence, they were subject to suspension or expulsion from school. However, these sanctions did not prevent them from continuing to use drugs. The pleasure obtained from drug use is a solution to escape from the uncomfortable conditions experienced by all participants at the moment. Once the addicts feel the positive effects of drug use, the cycle of seeking, using, and feeling pleasure repeats itself to uncontrollable use at the age of 17–18, when they are in high school or the first semester of college. All participants have made efforts to discontinue drug use. AG and MA underwent treatment voluntarily to relieve physical pain due to withdrawal symptoms. SG quit after being forced by his family to join the rehabilitation program, whereas PR and BB underwent rehabilitation according to court orders.

AG, MA, and PR have stopped using drugs for over ten years and have already reached the Maintenance Stage (DMR). In comparison, BB and SG are still in Stage 4 (Medium Recovery), characterized by a more stable lifestyle and a desire to return to real life, such as going to school, working, improving relationships with family, finding a life partner, and interacting with other people and a wider community. Especially in the case of SG, the characteristics of self-blame and cannot accept life history as an addict, as well as self-isolation from high-risk environments, still exist.

All the participants reported that they had tried to stop taking drugs, but then they experienced repeated relapses 2–6 times before finally reaching recovery and living a clean and sober life.

### 3.4. The Meaning-Making Process from ACE to Drugs Addiction and Recovery

3.4.1. Life before Drug Use

This study found that all participants experienced stressful events in their first 18 years of life and other significant crises before drug use. ACE scores ranged from 1 to 8 (Table 4) and impacted participants' beliefs about themselves and their perception of their existence. For example, "*I can say that since childhood I have been a lost child*" (AG), and his perception of life in response to experiences of being bullied during his school years was that "*the environment at that time was ruthless . . . not friendly*" (AG). BB described his relationship with his father as "*my father rarely talked to me, he talked to me only when he was angry... he often shouted at me, 'You are stupid! A useless child!'*" (BB) and his belief about himself was "*I am not wanted, not important and useless*"(BB). MA described his childhood experience: "*I need to be active doing things to get attention, but, in the end, I always make mistakes that make my parents angry and fight with each other. I felt guilty for doing bad things in the past, and it was not human behavior, it was like an animal*"(MA). PR who experienced emotional neglect expressed their condition during childhood "*I was a lonely child, I needed friends so badly, but going out to make friends was not allowed, in the end, I am not used to going out and had anxiety in a social environment*"(PR). SG believed that his background was problematic since his early life, revealing his childhood experience "*I was sick of being equated with and labeled after my father . . . I thought in the first place I never chose to be born to a drug addict father, a broken home family, and all kinds of things. In the end, I thought that maybe this is my identity, yes, becoming an addict, a criminal is my faith, I was supposed to do something like my father did*" (SG).

The initial appraisal of the ACEs is a critical starting point for the meaning-making process in later life and it can be understood as an implicit meaning [32]. In this study, various ways of appraisal are displayed and clearly understood through the initial attribution of the reasons for the occurrence, perceptions of the level of control, and their implications for the participants' future condition. Therefore, family background is vital in forming the initial implicit meanings of ACE. In this study, almost all participants experienced emotional neglect from their families. The appraised event meaning has several determining factors, such as the extent to which the event is threatening and controllable, the initial attribution of why the event occurred, and its implications for a person's future [33,34]. When participants were asked, "how do you view yourself concerning early life events?", their answers included, "I was a lost child," "I was weak, and the social environment was cruel," and "I was not wanted, and I am not important," "I was a bad boy, and my parents often fight because of it," "I was a lonely child, but my parents spoiled me with luxury," "my life has been wrong from the start," and "I always have problems."

It is not surprising that all participants reported that many childhood events were distressing since the occurrence of the incident continues. This condition then encourages participants to adapt and try to overcome the situation. Growing up, they made several efforts, such as stealing and hanging out with stronger groups, looking for ways to earn money by being drug couriers, buying friends, and other self-defense behaviors. The Meaning-Making Model describes their attempts to adjust to distressing situations as unconstructive.

### 3.4.2. Using Drugs as Adjustment to Discrepancy between Global Meaning and Situational Meaning

Global meaning is assumed to be constructed early in life and modified based on personal experiences, but global meaning nonetheless appears to influence individuals' thoughts, actions, and emotions [17,35]. Based on this explanation, family and environmental history construct participants' global meaning. At the same time, situational meaning refers to the meaning in the context of specific environmental encounters. Situational meaning thus begins with the occurrence of a potentially stressful event. It describes a series of ongoing processes and outcomes, including assigning meaning to the event (apprised meaning), determining the discrepancies between appraised meaning and global meaning, meaning-making, meaning-made, and adjustment to the event [17] (see Figure 1).

The appraised meaning of the events contained participants' attributions to ACE and other life crises and their ability to control perceived negative situations. The appraised meaning of the events contained participants' attributions to ACE and other life crises and their ability to control perceived negative situations. Stressful events during childhood are seen as threatening situations beyond their ability to control them. These stressful situations were interpreted as a condition that is not in line with expectations and causes distress. In the Meaning Making Model, distress arises because there are discrepancies between the assumptive world that acquire Global Meaning (GM) and the ACEs' appraised situational meaning (SM). In addition to ACEs, other life crises that occurred throughout periods of drug addiction also underwent similar meaning-making processes.

In answering questions about the participant's beliefs and emotions during childhood, we captured negative expressions and self-presentations, such as "I was a lost child."; "I am weird."; "I am weak."; "I was not important."; "I was not accepted."; "I was a naughty child.", "I was a stubborn boy."; "I was an anxious child."; "I always make mistakes."; "I am not confident."; "I truly need a friend." and "I am an unruly child." These expressions showed that all participants had low self-esteem regarding adverse childhood experiences.

Participants used drugs to adjust to ACE-related maladaptive stress by reducing the discrepancies between appraised situational meaning (ASM) and global meaning (GM). During periods of drug use, phrases that reflect feelings of self-worth or high self-esteem appeared, such as "I am the best, the most stubborn, the dumbest, and the smartest."; "I am known as an active, smart, and stubborn child."; "I feel accepted and needed by my

friends."; "I am the coolest child in my circle.", "I am stubborn, but I am popular among the addict's friends."; and "I am a creative child to overcome difficulties."

These expressions are positive judgments about self and are considered false positive beliefs. During active use, appraisals that were not by reality and false-positive beliefs emerged, such as "Drugs have saved my life. If there had not been drugs, I would have died a long time ago because I would not have been able to cope with my loneliness", "After taking drugs . . . I was sane . . . When I did not take drugs, I was insane... During a drug withdrawal, there were physiological changes, which made me insane". This strategy is a non-constructive meaning-making because, in the long term, it potentially creates new problems that will widen the previous discrepancies.

3.4.3. Consequences of Addiction and Life Crisis and Depressive Symptoms

Drug use harms a person's life. During drug use, all participants received harsh sanctions and were expelled many times from school, they also experienced conflicts with their families, and some got divorced because of their use. Two participants had severe injuries from motor/car accidents due to drunk driving. Some participants were involved in criminal acts, such as embezzling money and producing drugs to finance their addiction. They had been arrested and experienced detention by the police. Besides, they had severe health problems, such as being diagnosed with HIV and undergoing a nephrectomy due to kidney failure.

In addition to experiencing a crisis condition due to drug use, they also experienced a life crisis unrelated to drug use, such as the death of a parent that brought deep regret resulting in further suffering and ultimately encouraged further drug use. Critical events occurred continuously from time to time, and participants were exposed to many more crises until, at one moment, they experienced the lowest point in their life, where they felt helpless and gave up. Unbearable suffering as a result of drug use prompted three participants in this study to carry out suicide attempts. The expressions conveyed included: "after taking heroin, I always thought, this must be stopped . . . thinking that in the following day I got up early in the morning from sweating all over my body, had to look for drugs to cease these symptoms, had to go to remote places with risk of caught by the police... I could not stand anymore, then finally I wanted to commit suicide, this time I gave up, I hit rock bottom".

Using the BDI-II symptoms list, we identified some symptoms presented during drug use, such as feeling deep sadness, being punished, pessimism, worthlessness, poor concentration, hating oneself, suicidal thoughts, and attempting suicide. However, the BDI-II measurement found that only one participant met the criteria of borderline clinical depression, whereas another had mild mood disturbance. These two participants had the shortest recovery period (4 years and less).

Other participants also had a similar experience of hitting-the-rock-bottom, and this moment became a turning point in which there was a change in one's outlook on drugs, a wish to stop drug use, and awareness to seek help. In this study, the meaning-making process before reaching a turning point took a long time, between 8 and 21 years from the initial use. The following are some expressions that portray the occurrence of a turning point: "I just wanted to experience a real-life... I wanted to feel how to live a real life. A real life could be fun, I wanted to survive . . . ", "Since I was diagnosed with HIV, I could not live carelessly anymore . . . I was committed to living a straight life, and being useful at least to others, to my employer . . . that is all I wanted". The hitting rock-bottom episode initiated a more adaptive meaning-making process. Family and drug addiction counselors are essential in assisting and guiding the participants in finding constructive meaning (meaning made).

### 3.4.4. Turning Points and Recovery from Addiction (Meaning Made)

The critical result to note here is that the five participants experienced the lowest conditions in their lives while being addicts, perceived as hitting rock bottom, and considered a turning point for seeking and receiving treatment. The condition of severe distress initiated the meaning-making process, which later led them to the recovery path. With three participants, turning points moments happened after they failed to make a suicide attempt by using an excessive dose of drugs. In one case, the turning point occurred after he was diagnosed with HIV and, at the same time, was involved with a legal case.

At the time this research was conducted, AG, MA, and PR were already in the 6th Recovery Stage (Maintenance phase) of the Developmental Model of Recovery [1]. The characteristics of Maintenance are that there has been a process of continuing the growth and development of life, such as being able to overcome the transition period to become an adult, overcoming problems in daily life, and defending oneself so as not to relapse for a certain period. They stopped using drugs for more than ten years. In this study, BB and SG, who had been abstinent for four years, were still in Stage 4 (Medium Recovery), with more stable lifestyle characteristics and a desire to return to real life, such as going to school, working, and improving relationships with family. However, there were still characteristics of self-blame and cannot accept their identity as an addict, and they still had the tendency to withdraw from environments considered high risk.

Finding meaning in life occurs through an iterative process and is not linear. GM's alignment with SM goes through a series of stressful experiences. Based on experiences that occurred in five participants of this study, the more stressful experiences that needed to be processed constructively occurred, the more repetitive cycles of processes needed. If the stressful experience does not lead to the acquisition of constructive meaning, then this experience piles up with new stressful experiences, and so on, until all negative experiences are perceived and interpreted in harmony. The journey of finding meaning from non-constructive to constructive meaning occurs in several ways, namely automatic versus deliberate processes, accommodation versus assimilation, and searching for comprehensibility versus searching for significant and cognitive versus emotional processes.

From the exploration of participants' addiction experiences, the researcher concluded that addicts complete their recovery when they achieve global goals formed in childhood and can construct positive-false beliefs and subjective meaning or purpose in a way that GM and SM were in alignment. For example, in the case of AG, who had a global goal of "*I had to survive*" when facing bullying in childhood, later on in the Final Stage (6th) of DMR, ended up with a new perception of the global goal and perception of "*I survived.*" That was when quitting drugs was perceived as one of his outstanding achievements in life. He finally graduated with a bachelor's degree and was appointed as a permanent employee. He has been in recovery for 11 years.

All the participants underwent drug addiction therapy and rehabilitation after hitting rock bottom. As time went by and after experiencing a prolonged life crisis, they reached a new understanding of life and their drug addiction. This study found that the longer the recovery process, the more deeply the participants gained an insight that the world can be understood and that their lives were valuable. In this study, the meaning made is the result of meaning processes gained through processing early adverse experiences, the negative consequences due to drug use, and other life crises to the moment of turning point.

After going through many circles of the meaning-making process, participants made some new constructive meanings to various negative experiences during life as an addict. Meaning made reported by the participants of this study was not only about acquiring meaning in life as the goal of full recovery, but also throughout the process of addiction experience, including the changing of appraised event meaning of ACE and the reasons for its occurrence, changes in perceptions of self, and the meaning of drugs and drug use itself, and changes of the individual's view of the world.

The following are some illustrations of meaning made: (1) Feeling that something makes sense: "*I realized that my over-confidence caused me to slip and relapse.*"; "*Using drugs to*

*increase prestige is very risky.*"; (2) Accepting a situation: "*I began to realize the fact that life is not easy, but I have to live, that is life.*"; (3) Reattribution/having a causal understanding: "*In the past, drugs saved me. Now that I am diagnosed with HIV, if I do not want to die in vain, I must not take drugs . . . I still want to live the rest of my life as a useful person.*"; (4) The existence of the perception of growth or a positive change in life: "*I wanted to restore what I had lost... I wanted to take care of the things I had wasted, which gave me a feeling of satisfaction. Now, I feel my life and myself were still useful for others.*"; (5) Changing identity: "*In the past, I was a perfidious son, and I behaved like an 'animal'" . . . . "now I am 'more human' . . . *"; (6) Reassessing meaning of the stressor: "*I think it is hard to blame my biological father or mother. Maybe they did that because the situation forced them to do those things. They had not yet matured when they had a child.*"; (7) Changing global belief: "*In the past, drugs saved me. Now that I have been infected with HIV, to survive, I must avoid taking drugs.*"; (8) Changing global purposes: "*I am becoming increasingly aware that the world is full of dangers. In the world, there are many bad people and bad things, but I think my purpose in life is, at least, to make this world less messy or make me less sick. That is how I see the world now.*"; and (9) Restoring/changing meaning in life: "*Yes, in the past, it was impossible to go out without drugs. I needed drugs badly. Without drugs, something was missing in me, that is, self-confidence. Now, everything is beautiful without drugs. Now, the world is more colorful, right.*".

The participants achieved meaning in life in different ways. However, at least they experienced growth or a positive change in life, which enabled them to change their identities from addicts to recovering addicts. Situations that describe recovery from drug addiction go hand in hand with trauma recovery. The more stressful events experienced, the more meaning made should be achieved. Not all of the participants in this study had succeeded in achieving meaning-made of all the stressful events they experienced. These facts explain why some participants have reached the final stage of recovery in DMR, and some are still in the intermediate stage. Acquiring complete constructive meaning for all stressful events may characterize full recovery.

### 3.5. The Dynamics of Drug Addiction Recovery from the Perspective of the Meaning Making Model (Park, 2010)

An analysis of the experiences of the participants undergoing recovery shows that the dynamics of using drugs for the first time, experiencing drug addiction, and being in drug addiction recovery follows the following pattern: (1) the trauma from adverse childhood experiences (ACEs) brought forth distress, which led to discrepancies between GM and SM and the emergence of a subjective sense of purpose; (2) the discrepancies between GM and SM were reconciled by applying a non-constructive adaptation strategy, that is, taking drugs and other high-risk behaviors; (3) the use of drugs in the short term reduced the distress, but in the long term it brought forth more significant problems and additional distress; (4) when the problems became more significant, the participants hit-the-rock-bottom and had symptoms of depression; (5) depression became a turning point that opened up an opportunity for seeking help and initiating more constructive meaning-making; (6) drug addiction recovery was started by replacing the previous appraised meaning of the critical situation with the one that was adaptive and constructive, that is, restoring the wrong meaning made (positive-false belief) and then finding new meaning in life (Meaning Made); and (7) full recovery from drug addiction acquired when the achieved meaning in life was in line with the sense of purpose that had been developed based on the appraised meaning of the ACEs and life crises. Figure 2 is a chart showing the process of drug addiction recovery from the perspective of the Meaning Making Model [17].

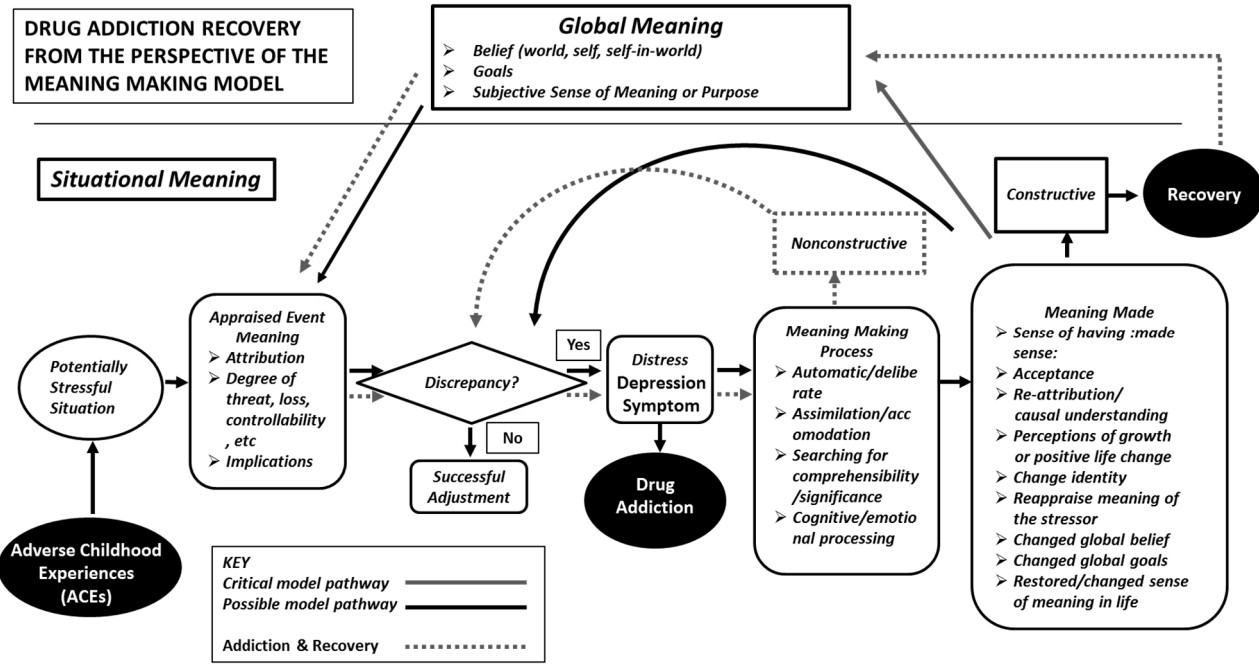

**Figure 2.** The meaning making model [17] in drug addiction recovery.

## 4. Discussion

Based on the results, the participants' drug use behavior lasted for 10 to 17 years before they realized that the drug use had caused problems, and the desire to recover from drug use emerged. Some participants' recovery started in their twenties, whereas others were in their thirties. Several factors, including cognitive abilities, the pressure to drug use, drug access, intervention from the family to stop drug use, and how quickly the participants experience a turning point or hitting rock bottom, influenced the above condition.

The results of ACE-related studies on participants confirmed the findings of previous studies that ACE has a strong relationship with drug addiction, where people with an ACE score of 4 and above are 11 times more likely to become addicted to drugs [23,25,30,36]. In this case, three participants with ACE scores below four become drug addicts. This fact explains that adverse experiences affect a person's vulnerability to using drugs later in life [29]. In addition, this fact can also explain that difficult experiences in childhood (such as neglection, abuse, or having an addicted parent/caregiver) and family environmental situations (such as parent-child interactions and parenting patterns) can be predictors of drug addiction [37–39].

The participants' drug use started from late childhood to early adolescence, which is the phase of searching for self-identity [40]. Regarding participants' perceptions of their identity and life, the results of this study give a similar picture to the results of a previous study, in which addicts had a negative perception of self-identity and life before they started using drugs [4]. Through a constructive meaning-making process, participants changed their perception of their identity from one based on false positive beliefs to a more realistic identity. The idea that drug use is a way to find one's identity [5] can be justified here through the participants' descriptions.

Regarding the role of depression in the process of recovery, the study's results confirm the opinion that distress may not only become a factor underlying drug use but also initiate a search for new sources of meaning to replace previously appraised event meanings [5,20]. As reported, the situation in which the participants hit rock bottom became a turning point for them to seek help and reappraise the improper meaning they had made. The depressive episode became a strong motivation for them to start the process of recovery from drug addiction.

The meaning-making process occurred continuously and repeatedly (non-linear) through critical experiences and repeated relapses until complete recovery. Relapses would not stop until the root of the problem causing the use of drugs, i.e., the trauma from ACEs, was addressed. This finding is in line with the opinion that most drug addicts have experienced trauma, and if the addicts do not heal from the trauma, it will be difficult for them to recover from drug addiction [25,26]. The results of this study show that drug addiction recovery is in line with post-traumatic growth (PTG) in trauma recovery. The study also supports the opinion of Moulds and Bryant [19] that PTG among individuals experiencing trauma is related to their process of accommodation and changes in their global beliefs.

Concerning the meaning of traumatic experiences, the results of this study indicate that ACE can predict individual participants to recover if the individual has achieved a more productive meaning (meaning made) of the stressful events throughout life. In the context of understanding meaning in life as a construct, by analyzing the result of Searching of Meaning and Presence of Meaning in MLQ with the meaning-making process of all participants, it is understood that the Searching of Meaning and Presence of Meaning [15] equate with the acquisition of meaning-making process and meaning made in Meaning Making Model [17]. The Meaning Making Model [17] is suitable for explaining the meaning-making process in a single stressful event. However, this model is also suitable to be applied to explain the dynamic of multiple stressful events in life, as proposed in Figure 2.

## 5. Conclusions

This study may suggest that the Meaning Making Model [17] can be used to explain how an individual develops drug addiction and how recovery from drug addiction occurs. In addition, the study has produced some findings related to the implementation of the Meaning Making Model [15] in the investigation of drug addiction and recovery from it, namely:

- Drug use is a maladaptive and non-constructive strategy to overcome distress due to ACEs and other life crises.
- During active drug use, a false-positive belief about drugs and self-concept caused drug use to be maintained and increased.
- The drug addiction recovery process was initiated with a change from maladaptive and non-constructive meaning-making to adaptive and constructive meaning-making after the participants hit rock bottom, resulting from repeated critical experiences.
- Relapses continuously occur if the meaning of the root of the problem that caused drug use, i.e., ACEs, had not yet been appraised constructively, although addicts have already found constructive meaning about drug use.
- All the participants, through the process of meaning-making, changed their self-identity from one based on a false-positive belief to a more realistic self-concept.

The findings of this study show the importance of finding meaning in life that is in line with trauma recovery during drug addiction recovery. Therefore, studies on the recovery journeys of former drug addicts who have completed drug addiction recovery are as critical as studies on developmental trajectories of drug addictive behaviors, especially traumatic experiences that may underlie drug use. In other words, the whole sequence of drug addiction—before and during drug use and during drug addiction recovery—needs to be considered.

The strength of this study is that an in-depth qualitative approach could explore the dynamics of the relationship between ACE and drug addiction. In addition, this research can better understand the journey of recovery through the acquisition of meaning in life. The context of this study includes the developmental trajectory of drug addiction behavior and the recovery path before drug use, during active drug use, and until achieving recovery.

Methodologically, this research has some limitations, such as limited cases and the data obtained relying more on a retrospective piece of information, so there may still be important information that is not conveyed. In connection with research topics classified as

sensitive (trauma and addiction), there is a tendency to avoid talking about and recalling traumatic events. Consequently, there is a tendency to avoid recalling them, or a "false memory syndrome" occurs. Additionally, there is a possibility that participants have difficulty remembering traumatic experiences, so the information conveyed does not fully reflect the condition. Some participants need a relatively long time to build rapport and to reveal in-depth information. Online data collection risks the emergence of technical problems and difficulties in controlling the conditions of the interview from environmental disturbances in which the participants are located. Based on these limitations, similar research needs to pay more attention to rapport development, which takes longer. Data collection is best done face to face.

Further research that focuses on a cross-sectional analysis of meaning-making in the recovery stages can be recommended. In addition, this intervention is expected to accelerate recovery and prevent relapse. The result of this study may enrich the treatment modality in drug addiction rehabilitation programs, especially using the spiritual approach as part of the BPSS Model. Meaning therapy may shorten the period of both drug addiction recovery and trauma recovery.

**Author Contributions:** Conceptualization, data curation, formal analysis, methodology, writing—original draft, T.I.; Formal analysis and writing—review and editing, Z.L.D.; Supervision, W.W.M. and I.I. All authors have read and agreed to the published version of the manuscript.

**Funding:** This research received no external funding.

**Institutional Review Board Statement:** The Ethics Committee of The Faculty of Psychology, the University of Indonesia, approved the study (364/FPsi.Komite Etik/PDP.04.00/2018).

**Informed Consent Statement:** Informed consent from all subjects involved in the study is available.

**Data Availability Statement:** Data is contained within the article.

**Conflicts of Interest:** The authors declare no conflict of interest.

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
