# Peer review of "Meaning-Making among Drug Addicts during Drug Addiction Recovery from the Perspective of the Meaning-Making Model"

_psych, doi:10.3390/psych4030045_

Round 1
Reviewer 1 Report
The authors have examined meaning-making in drug addiction based on the life stories of 5 recovered addicts. This paper has the potential to offer new insights into how people with drug addiction have developed their meaning in life. As it stands, however, this does not become sufficiently clear to me. Please see the attachment for more specific questions and suggestions.
My general recommendation for the authors is to look more critically at their data and at the meaning making model. From the current description it does not become clear what the global meanings and the situational meanings of the participants were, how these were in conflict with each other and what the participants did to resolve these conflicts and what meanings were made.
In addition, it is unclear what the relationship of the authors is to the participants; the research process sound very intense and deals with very sensitive information, so more information is needed on how participants were recruited, what the relationship with the interviewers was and how the well-being of participants was secured.

Author Response
Reviewer 1
Review Report Form
Open Review
( ) I would not like to sign my review report
(x) I would like to sign my review report
English language and style
( ) Extensive editing of English language and style required
(x) Moderate English changes required
( ) English language and style are fine/minor spell check required
( ) I don't feel qualified to judge about the English language and style
Regarding English language and style, we double-checked all sentences and then edited by paraphrasing and correcting grammatical errors in all parts, from the abstract to the conclusion sections.
Yes Can be improved Must be improved Not applicable
Is the content succinctly described and contextualized with respect to previous and present theoretical background and empirical research (if applicable) on the topic?
(x) ( ) ( ) ( )
Are all the cited references relevant to the research?
(x) ( ) ( ) ( )
Are the research design, questions, hypotheses and methods clearly stated?
( ) ( ) (x) ( )
Are the arguments and discussion of findings coherent, balanced and compelling?
( ) ( ) (x) ( )
For empirical research, are the results clearly presented?
( ) ( ) (x) ( )
Is the article adequately referenced?
(x) ( ) ( ) ( )
Are the conclusions thoroughly supported by the results presented in the article or referenced in secondary literature?
( ) ( ) (x) ( )
Comments and Suggestions for Authors
- The authors have examined meaning-making in drug addiction based on the life stories of 5 recovered addicts. This paper has the potential to offer new insights into how people with drug addiction have developed their meaning in life. As it stands, however, this does not become sufficiently clear to me. Please see the attachment for more specific questions and suggestions.
Thank you for sending us the detailed feedback on the previous manuscript. By carefully reading and reviewing the comments, we realized that more explanation, elaboration, and changes are absolutely needed, not only in the introduction section, but also in the result, discussion, and conclusion sections. Below are the points of revision in response specifically to comments in the previous manuscript:
- Page 2, Paragraph 1, Line 46-56, we changed the key questions and reformulated all explanations about meaning and spirituality.
- Page 2, Paragraph 3, Line 63-68, we paraphrased and rearranged almost the whole paragraph to correct the repetitive and incomplete sentences.
- Page 2, Paragraph 4, Line 69-80, we added more information to specify the nature of the found relationships.
- Page 2, Paragraph 5, Line 85-88, we added another theory that is strongly related to the meaning-making model with the aim to give more explanation about the meaning-making process.
- Page 3, Paragraph 1, Line 104-123, we made this paragraph to describe more the positive nature of meaning made, specifically a change in global beliefs and goals, in helping individuals not only to resolve the discrepancy but also to enhance well-being. Here, we also put some example cases to illustrate the explanation.
- Page 3, Paragraph 3-4, Line 138-149, we specified the accommodations and the new beliefs and goals by adding an example case of traumatic experience and PTG.My general recommendation for the authors is to look more critically at their data and at the meaning making model. From the current description it does not become clear what the global meanings and the situational meanings of the participants were, how these were in conflict with each other and what the participants did to resolve these conflicts and what meanings were made.
After closely looking at and reanalyzing our data and exploring the meaning-making model more, we decided to make enormous changes in the result section, mainly by changing the presenting result from the meaning-making model to the drug addiction and recovery standpoints. The changes in the result structure and presentation, as well as in the content can be found below:
- We adjusted the title of the sub-sections on Page 6 Line 260, Page 7 Line 272, Page 7 Line 292, Page 8 Line 345 and 346, Page 10 Line 394, Page 11 Line 440, Page 12 Line 482.
- We added Tables along with the adjustment to the Tables’ numbers and captions on Page 6 Line 270 and Page 7 Line 290.
- Page 6, Sub-Section 3.1, we focused the explanation here on the demographic characteristic of the participants in terms of age, gender, initial use, length of drug use, and recovery period.
- Page 7, Sub-Section 3.2, we added this section to include the assessment results of ACE, Meaning in Life, and Depression Symptoms by applying the Adverse Childhood Experience Questionnaire (ACE), Meaning in Life Questionnaire (MLQ), and Beck's Depression Symptoms (BDI)-II. We used these assessment results to complement and support the analyses of the main result from the in-depth interview.
- Page 7-8, Sub-Section 3.3, we completed the participants’ background by adding their drug addiction history (Line 313-343). We also paraphrased the statement in Line 308-312 to express about abusive family in a more neutral way as well as removed all information about the location to ensure further anonymization.
- We deleted the statement about how ACE would be appraised as the experience that initiate the discrepancy between GM-SM. We agree with Reviewer 1 that it is too soon to draw conclusions here since we should describe first the global and situational meaning of the participants to know it clearly.
- To complete the description of the point above, we added the explanation about the implicit meaning of ACE that is predicted to rise to a distress situation as, according to Park (2010), a consequence of the discrepancy between GM-SM on Page 10, Line 372-393. Specifically, in Line 382-386 we also added some quotes from the participants that would more subjectively express appraised events meaning related to ACE.
- Page 11, Line 424-439, we paraphrased and revised the explanation about how we analyzed that drug use of the participants could be seen as their adjustment to ACE-related maladaptive stress. We also added some quotes here that represent the participants’ self-esteem and self-worth that could be seen as the false-positive beliefs, which initiate the discrepancy or conflict between global meaning and appraised situational meaning.
- We added the description of other life crises, including the quotes that expressed the unbearable suffering and hit-the-rock-bottom of the participants in Sub-Section 3.4.3, Page 11, Line 450-461.
- To enrich the description of the hit-the-rock-bottom of the participants, we also added more explanation as well as quotes on Page 11-12, Line 469-480.
- Section 3.4.4 Page 12-13, we reconstructed the explanation of meaning made by giving a short overview of the process of hit-the-rock-bottom and turning points (Paragraph 1, Line 484-491). Furthermore, we added the DMR theory, to strengthen our focus on drug addiction and recovery (Paragraph 2, Line 492-502). Paragraphs 3-7 Line 503-567, we elaborated the explanation of the meaning-making process, including ACE as appraised situational meaning and addiction and recovery journey. In the last paragraph, we explicitly revised the quotes that represent each of the nine categories of meaning made.
- Page 16, Paragraph 2-4, Line 684-709, we added our critical reflection on the strengths and limitations of the study and propose recommendations for further research.
In addition, it is unclear what the relationship of the authors is to the participants; the research process sound very intense and deals with very sensitive information, so more information is needed on how participants were recruited, what the relationship with the interviewers was and how the well-being of participants was secured.
We notice that these comments are related to the research method, mainly about the participants and the research procedures. With this regard, we added some more information below:
- We completed the information about the participants' criteria by adding the selection and data collection process on Page 4-5, Line 174-209. We also adjusted the title of this section.
- We added Table 1 which consists of the Semi-Structure Interview Guide in Line 226 followed by the data collection and documentation procedures on Page 5, Line 226-239.
- The explanation about assessment of depression symptoms was explained in the result section (Page 7, Line 283-288).

Reviewer 2 Report
With a sample this small any conclusions must be tentative. A serious editing for grammar and style is in order. There was a pattern of using a reference number at the start of a sentence (see line 63 as an example) this is improper. In broader terms you methodology and subsequent conclusions have a disconnect and logical error. using a preconceived model to interview 5 subjects and then claim the model was supported by a collection of random quotes is suspect. One could argue that you found exactly what you were looking for because that is exactly what you planned. To convince me I would require a more objective analysis.
Author Response
Reviewer 2
Review Report Form
Open Review
( ) I would not like to sign my review report
(x) I would like to sign my review report
English language and style
(x) Extensive editing of English language and style required
( ) Moderate English changes required
( ) English language and style are fine/minor spell check required
( ) I don't feel qualified to judge about the English language and style
Regarding the issue of English language and style, we double-checked all sentences and then edited by paraphrasing and correcting grammatical errors in all parts, from the abstract to the conclusion
Yes Can be improved Must be improved Not applicable
Is the content succinctly described and contextualized with respect to previous and present theoretical background and empirical research (if applicable) on the topic?
( ) ( ) (x) ( )
Are all the cited references relevant to the research?
( ) (x) ( ) ( )
Are the research design, questions, hypotheses and methods clearly stated?
(x) ( ) ( ) ( )
Are the arguments and discussion of findings coherent, balanced and compelling?
( ) ( ) (x) ( )
For empirical research, are the results clearly presented?
(x) ( ) ( ) ( )
Is the article adequately referenced?
(x) ( ) ( ) ( )
Are the conclusions thoroughly supported by the results presented in the article or referenced in secondary literature?
( ) ( ) (x) ( )
Comments and Suggestions for Authors
With a sample this small any conclusions must be tentative
The qualitative approach using in-depth interviews applied here aims to draw conclusions about a dynamic and in-depth process. Therefore, in terms of the number of participants, the participants who were involved from the beginning to the end of the interview were indeed few. However, in terms of data, we obtained so much in-depth data, unique to each participant but at the same time, there is a similar pattern in terms of how they find meaning in life during their life journey from before using drugs, during being addicts, to achieving drug recovery. Of course, with this qualitative approach, generalizable conclusions are not the aim of our study. Because of this, we agree that more consistent conclusions require more data that is supported by quantitative measurement results.
A serious editing for grammar and style is in order.
Thank you for the feedback. We have tried to edit grammar, structure, and language expression in this revised manuscript. However, we realize that formal English will be always challenging for us. Concerning this issue, we will always be open to learning and getting help from any proofreader along the process of publication.
There was a pattern of using a reference number at the start of a sentence (see line 63 as an example) this is improper.
We have edited this error by viewing and following published articles using the reference writing system as implemented here. However, we are very open to being advised if there are still errors in this revised manuscript. Of course, we will make corrections again later, especially if we get the opportunity to publish our article in the Psych journal.
In broader terms your methodology and subsequent conclusions have a disconnect and logical error. using a preconceived model to interview 5 subjects and then claim the model was supported by a collection of random quotes is suspect.
We really appreciate critical feedback on this matter. With this critical response, we were inspired to rethink and review everything related to the data applied in a model.
Basically, the meaning-making model used in this study is intended to describe the dynamics of the meaning-making process in addicts from before using drugs, during drug action, until recovering. Therefore, this model is used as a reference in making interview guidelines (Page 6, Line 226) with the aim of obtaining complete data from all participants based on the theoretical concepts of the model (Result Section, Sub-Section 3.4, Page 6-14, Line 245-595) . In this case, the concept is explained in the context of drug addiction which is another focus of this research.
By using an in-depth interview technique that uses thematic analysis, the data that we report here is the result of the coding process for in-depth answers from participants regarding the complex and long process of drug recovery. The quotes that we write in the manuscript are examples of participant expressions that we analyze and then choose because they represent what we are discussing. In line with this, the conclusions we draw are more directed at how the dynamics of this model occur in addicted participants in this study, not to get an answer whether this model is supported or not by our data.
One could argue that you found exactly what you were looking for because that is exactly what you planned. To convince me I would require a more objective analysis.
Thanks for this constructive suggestion. In the results section (Page 6-13), discussion (Page 14-15), and Conclusion (Page 15-16 ) we try to draw the existing data, then interpret it based on the theoretical concepts we use in this study.
Quotes taken from verbatim data by following standard procedures from a qualitative research, then interpreted using a scientific theoretical framework, accompanied by relevant supporting scientific literature is a necessary condition for reporting research results objectively and scientifically.
In this revised version, we have tried to improve the structure of writing, explanations, and also the selection of examples of quotes (which are direct expressions from participants). We sincerely hope that our efforts will convince you that our analysis is objective.

Round 2
Reviewer 2 Report
This is a much improved article. As I read it now, especially with the ACE material, I see deeper value and possibilities.
Author Response
DETAIL REVISION
Academic Editor Notes
The article is interesting and adds to the discussion of the role of spirituality in the recovery from drug addiction in individuals with traumatic life experiences.
The writing still needs to be improved to eliminate overstatements and typos such as:
- a) phrase to be corrected
.. they realized that they realized that ...the drug use had...
We have fixed the error on Page 14, Line 582
- b) the phrase ...the idea that....drug use is a way to find meaning....is justified by the participant's responses. it is incorrect,
In the Editor's opinion, the fact that the study participants managed to find meaning after the use of drugs absolutely does not support the idea that drug use is a way to find meaning.in life. Most addicts did not find meaning but premature death?
The study participants managed to find a constructive meaning during the recovery process. The drug addiction recovery process was initiated with a change from maladaptive and non-constructive meaning-making to adaptive and constructive meaning-making after the participants hit rock bottom, resulting from repeated critical experiences. The meaning-making process occurred continuously and repeatedly (non-linear) until achieved full recovery.
The perspective of Existentialism explains drug addiction as a response to a life that lacks personal meaning caused by the addict's hampered ability to choose an authentic and meaningful life. The fact that premature death happens among addicts occurs because of uncontrolled use, and this can be due to the belief that what makes them alive and happy is only drug use. Therefore, we may suggest the importance of facilitating the meaning-making process through guidance and support for recovering addicts to prevent premature death.
- c) In conclusion
the authors said that the study has proved...
It is the editor's opinion that a retrospective case study with a reduced number of participants involved and unintentional bias induced by the selection of the participants may suggest..but not prove any theory or idea..please correct.
We have corrected it as below:
This study may suggest that the Meaning Making Model can be used to explain how an individual develops drug addiction and how recovery from drug addiction occurs (Page 15, Line 636)
Reviewer 2
Open Review
( ) I would not like to sign my review report
(x) I would like to sign my review report
English language and style
( ) Extensive editing of English language and style required
( ) Moderate English changes required
(x) English language and style are fine/minor spell check required
( ) I don't feel qualified to judge about the English language and style
Yes Can be improved Must be improved Not applicable
Is the content succinctly described and contextualized with respect to previous and present theoretical background and empirical research (if applicable) on the topic?
(x) ( ) ( ) ( )
Are all the cited references relevant to the research?
(x) ( ) ( ) ( )
Are the research design, questions, hypotheses and methods clearly stated?
(x) ( ) ( ) ( )
Are the arguments and discussion of findings coherent, balanced and compelling?
(x) ( ) ( ) ( )
For empirical research, are the results clearly presented?
(x) ( ) ( ) ( )
Is the article adequately referenced?
(x) ( ) ( ) ( )
Are the conclusions thoroughly supported by the results presented in the article or referenced in secondary literature?
(x) ( ) ( ) ( )
Comments and Suggestions for Authors
This is a much improved article. As I read it now, especially with the ACE material, I see deeper value and possibilities.
Thank you for the positive comment.
